# Genome-Wide Analysis of Gene Expression Noise Brought About by Transcriptional Regulation in *Pseudomonas aeruginosa*

Wenhui Chen,[a] Jinfeng Zhang,[a] Feixuan Li,[a] Congcong Wang,[c] Yuchen Zhang,[a] Aiguo Xia,[b] Lei Ni,[b] Fan Jin[b]

[a]Hefei National Research Center for Physical Sciences at the Microscale; Department of Polymer Science and Engineering, University of Science and Technology of China, Hefei, China
[b]CAS Key Laboratory of Quantitative Engineering Biology, Shenzhen Institute of Synthetic Biology, Shenzhen Institutes of Advanced Technology, Chinese Academy of Sciences, Shenzhen, China
[c]Department of Chemical Physics, University of Science and Technology of China, Hefei, China

Wenhui Chen, Jinfeng Zhang, and Feixuan Li contributed equally to this article.

**ABSTRACT** The part of expression noise that is brought about by transcriptional regulation (represented here as NTR) is an important criterion for estimating the regulatory mode of a gene. However, characterization of NTR is an under-explored area, and there is little knowledge regarding the genome-wide NTR in the model pathogen *Pseudomonas aeruginosa*. Here, with a library of dual-color transcriptional reporters, we estimated the NTR for over 90% of the promoters in *P. aeruginosa*. Most promoters exhibit low NTR, while 42 and 115 promoters with high NTR were screened out in the exponential and the stationary growth phases, respectively. Specifically, a rearrangement of NTR was found in promoters involved in amino acid metabolism when bacteria enter the exponential phase. In addition, during the stationary phase, high NTR was found in a wide range of iron-related promoters involving siderophore synthesis and heme uptake, ExsA-regulated promoters involving bacterial virulence, and FleQ-regulated promoters involving biofilm development. We also found a large-scale negative dependence of transcriptional regulation between high-NTR promoters belonging to different functional categories. Our findings offer a global view of transcriptional heterogeneity in *P. aeruginosa*.

**IMPORTANCE** The phenotypic diversity of *Pseudomonas aeruginosa* is frequently observed in research, suggesting that bacteria adopt strategies such as bet-hedging to survive ever-changing environments. Gene expression noise (GEN) is the major source of phenotypic diversity. Large GEN from transcriptional regulation (represented as NTR) represent an evolutionary necessity to maintain the copy number diversity of certain proteins in the population. Here, we provide a system-wide view of NTR in *P. aeruginosa* under nutrient-rich and stressed conditions. High NTR was found in genes involved in flagella biosynthesis and amino acid metabolism under both conditions. Specially, iron acquisition genes exhibited high NTR in the stressed condition, suggesting a great diversity of iron physiology in *P. aeruginosa*. We further revealed a global negative dependence of transcriptional regulation between those high-NTR genes under the stressed condition, suggesting a mutually exclusive relationship between different bacterial survival strategies.

**KEYWORDS** NTR, *Pseudomonas aeruginosa*, dual-color transcriptional reporter library, iron

Address correspondence to Lei Ni, lei.ni@siat.ac.cn, or Fan Jin, fan.jin@siat.ac.cn.

The authors declare no conflict of interest.

While noise in gene expression is inherent, it can also be modulated by transcriptional regulation. For example, bacteria can increase their gene expression noise via transcriptional regulation to adopt a bet-hedging strategy in ever-changing environments (1, 2). Transcription-initiated gene expression noise (TGEN) can be divided into two parts:

(i) stochastic noise arising from the near-Poisson behavior of transcription initiation, and (ii) noise arising from fluctuations in both the expression level and the promoter-binding state of upstream transcription factors (3–6), represented here as NTR. For a constitutive promoter whose transcription is unregulated by any transcriptional regulators, NTR is very near zero. In this instance, the difference in transcription-initiated gene expression noise between an interested promoter and a constitutive promoter with the same mean transcription level is NTR. Over billions of years of natural selection, NTR has been a part of gene expression noise that bacteria can control through the evolution and adjustment of specific regulatory networks. Thus, the NTR, rather than the total expression noise, reflects the underlying regulatory mode of a target gene (7–10). To date, excellent theories have been developed to understand the underlying logic of NTR regulation (3, 5, 9, 11), and finding genes with high NTR promises to uncover new mechanisms by which bacteria adapt to their habitats. However, difficulty in distinguishing NTR from the stochastic noise brought about by transcription in experiments hinders characterization of NTR, and genome-wide knowledge of NTR in bacteria is lacking. We aimed to characterize the NTR of gene expression at a systematic level.

The fluorescent protein-based transcriptional reporter is a commonly used tool to characterize the expression variability for a large number of genes (12–14). Expression noise can be estimated from the steady-state distribution of protein copy numbers in single cells (12–14). The intrinsic and extrinsic noise of gene expression can be further characterized with dual-color reporters, i.e., two identical promoters to drive the expression of two different fluorescent proteins in a single cell (15). Researchers also use dual-color reporters with two different promoters (a target promoter and a constitutive promoter) to calculate the global and promoter-specific intrinsic noise (16). In the latter case, NTR is contained in the promoter-specific intrinsic noise, which is greater than the pure intrinsic noise calculated from a dual-color reporter of identical target promoters. Thus, NTR can be estimated according to the deviation of promoter-specific intrinsic noise from the theoretical curve of pure intrinsic noise (5, 11). Here, we report single-cell global profiling of NTR using a library of dual-color transcriptional reporters for the model pathogen *Pseudomonas aeruginosa*. We screened out multiple promoters with high NTR in both the exponential and stationary growth phases, with functional categories containing amino acid metabolism, virulence, biofilm formation, efflux pumps, iron acquisition, etc. The high-NTR promoters found in this study provide a wealth of new clues for further study of transcriptional regulation in *P. aeruginosa*.

## RESULTS

**A dual-color transcriptional reporter library for *P. aeruginosa*.** To construct the library in the PAO1 strain, we used *sfGFP* as a quantifiable marker for the transcription of each candidate promoter and used constitutively expressed *cyOFP1* as an internal standard; both modules were cloned into the PUCP20 backbone (Fig. S1 in the supplemental material). The speculative promoter sequences of each open reading frame (ORF), including the entire intergenic region and part of the coding sequence of the upstream gene, were determined according to the PAO1 genome database. We designed 3,339 transcriptional reporters for all of the ORFs in PAO1, of which 3 failed during plasmid construction, 112 showed significant growth defects, and 199 exhibited a low expression level below our microscope's limit of detection (see Table S1 at https://figshare.com/articles/dataset/Figures/21266778). Thus, we finally obtained qualified expression data for promoters of 3,025 ORFs, accounting for 90.6% of all ORFs in PAO1. The reporter strains were grown in LB medium. The single-cell fluorescence of SfGFP and CyOFP1 for each reporter in both exponential and stationary growth phases was evaluated by microscopy. Fluorescent intensity was transformed to SfGFP or CyOFP1 numbers in single cells (see Materials and Methods). To assess the repeatability of our methods, we selected 100 reporters and repeated the data-collection procedure on two different days. R-squares for mean SfGFP expression during both growth phases reached 0.99, while R-squares for coefficients of variations (CV, ratio of standard

deviation to the mean) for SfGFP expression in the cell population during the exponential and stationary phases were 0.732 and 0.896, respectively (Fig. S2a to d). These findings imply that our experimental system has good reliability. To validate the effectiveness of the reporter library, we tested the response of several reporters under previously reported conditions (17–20) (Fig. S3). Ciprofloxacin treatment induced the expression of multiple SOS reporters in our library. *aceA* and *acsA* reporters exhibited significantly increased expression when acetate was used as the sole carbon source instead of succinate. In addition, we also repeated the phase variable cell-to-cell heterogeneous expression of *cupA1* when *mvaT* was deleted. These results demonstrate the reliability of our reporter library.

**Separation of gene expression noise and estimation of NTR.** To obtain an overview of gene expression noise for *P. aeruginosa*, we plotted the total noise ($\eta^2_{\text{total}}$) of gene expression against the mean expression level for all reporters (Fig. 1a and d). As previously reported (12–14), $\eta^2_{\text{total}}$ is strongly dependent on mean SfGFP expression level, with $\eta^2_{\text{total}}$ decreasing as SfGFP expression increases. Using expression data from the dual-color reporter system, the promoter-specific intrinsic noise ($\eta^2_{\text{int,promoter}}$) and promoter-independent extrinsic noise ($\eta^2_{\text{global}}$) can be determined as previously described (16):

$$\eta^2_{\text{total}} = \frac{<\delta G^2>}{<G>^2} = \frac{<G^2> - <G>^2}{<G>^2} \tag{1}$$

$$\eta^2_{\text{global}} = \frac{<\delta G \delta R>}{<G><R>} = \frac{<GR> - <G><R>}{<G><R>} \tag{2}$$

$$\eta^2_{\text{int,promoter}} = \eta^2_{\text{total}} - \eta^2_{\text{global}} \tag{3}$$

where $G$ and $R$ are the single-cell expression levels of SfGFP and CyOFP1, respectively. Brackets ($<,>$) represent the population average of the number of fluorescent proteins in single cells. $\eta^2_{\text{global}}$ measures the expression noise contributed by intracellular variations in global factors which is common to the target promoter and the constitutive J23102 promoter, such as RNA polymerase level, ribosome number, growth rate, and cell size. $\eta^2_{\text{global}}$ measures the expression noise contributed by the factors which independently affect the expression of the target promoter. $\eta^2_{\text{int,promoter}}$ consists of two parts: (i) expression noise arising from the inherent randomness in transcription and translation (represented here as pure intrinsic noise, $\eta^2_{\text{int,pure}} = (b + 1)/<n>$) and (ii) expression noise arising from fluctuations in both the expression level and the promoter-binding state of upstream transcription factors which are specific to the target promoter (represented here as NTR).

Because $\eta^2_{\text{global}}$ is promoter-independent, our results of 3,025 transcriptional reporters represent 3,025 repeated measurements of promoter-independent extrinsic noise. As expected, $\eta^2_{\text{global}}$ in the stationary phase exhibits a narrow distribution independent of the expression level, whereas $\eta^2_{\text{global}}$ in the exponential phase has a wider distribution (Fig. 1b and e), implying a greater experimental error for data in the exponential phase. The average $\eta^2_{\text{global}}$ is slightly larger in the stationary phase than in the exponential phase (0.128 to 0.118). In addition, $\eta^2_{\text{global}}$ exhibited a similar decrease with increased protein expression level, as observed for $\eta^2_{\text{total}}$ (Fig. 1c and f). We used the theoretical relationship $\eta^2_{\text{int,pure}} = (b + 1)/<n>$ to fit data (5), in which $b$ denotes the average number of proteins produced by one mRNA molecule and $<n>$ denotes the mean protein number per cell. This theoretical curve fits well with our $\eta^2_{\text{int,promoter}}$ data in both growth phases, indicating that most promoters are near constitutively expressed in *P. aeruginosa*. We obtained $b$ values of 43.1 in the exponential phase and 87.3 in the stationary phase, which is consistent with previous findings in *Escherichia coli* that the average mRNA lifetime in the stationary phase is about twice that in the exponential phase (7.8 min to 4.1 min) (21). In addition, compared with the data in the exponential phase, more discrete points away from the large

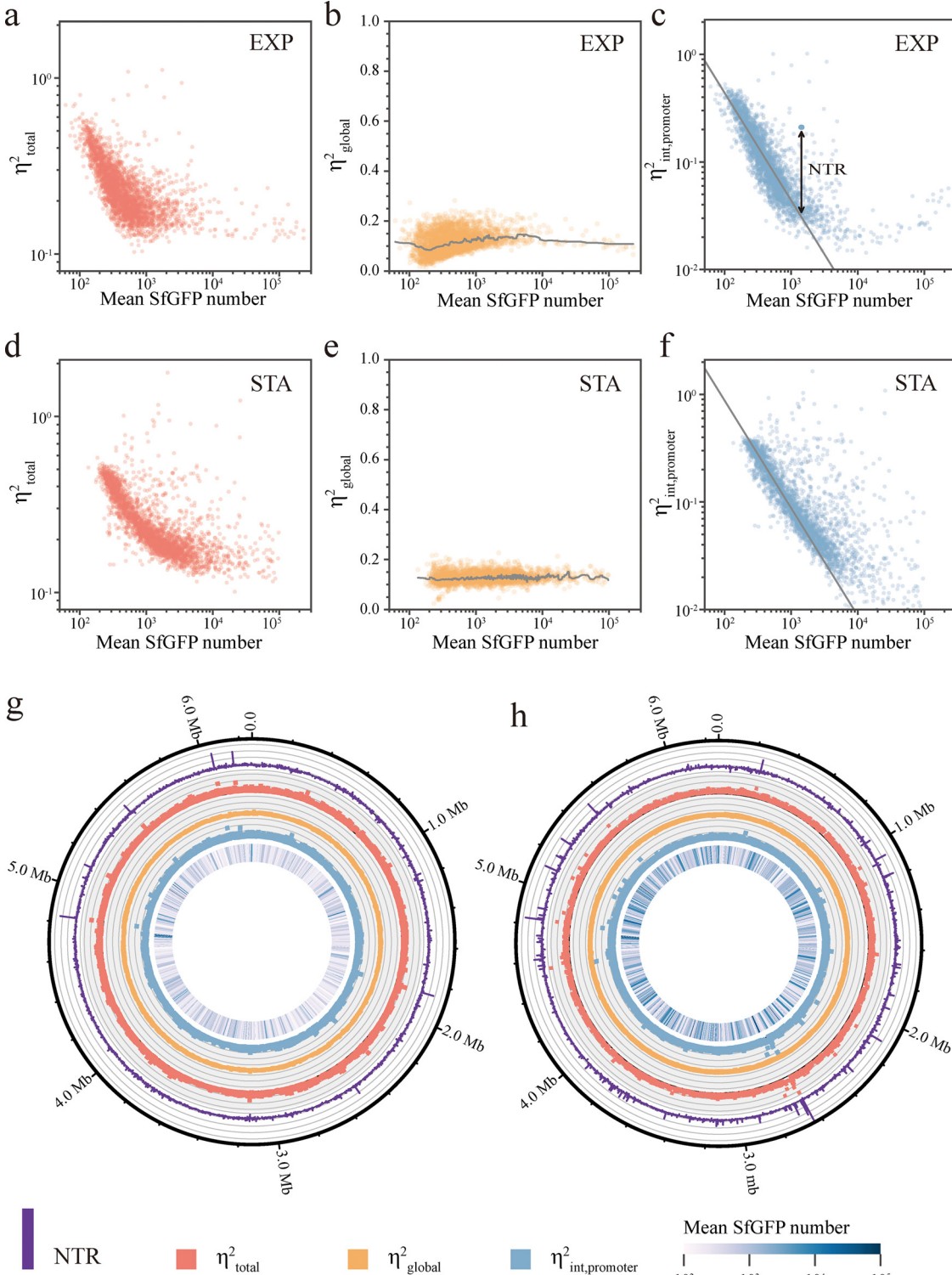

**FIG 1** Profiling of total noise, promoter-independent extrinsic noise, promoter-specific intrinsic noise and NTR (expression noise brought about by transcriptional regulation) (a to f) $\eta^2_{total}$ (a and d), $\eta^2_{global}$ (b and e), and $\eta^2_{int,promoter}$ (c and f) of promoter expression plotted against mean SfGFP number in cells for 3,025 reporter strains. EXP, exponential phase; STA, stationary phase. Running medians (smoothing window of 50) are shown in panels b and e (light gray line), fitting curves (light gray) with the relationship $\eta^2_{int} = (b + 1)/\langle n\rangle$ are shown in panels c and f. NTR is represented by the double-headed arrow in panel c. (g and h) Circos plot of promoter expression data in the exponential (g) and stationary (h) growth phases for 3,025 reporters in correspondence to their genomic locations. Traits from outer ring to inner ring: NTR (purple bar), $\eta^2_{total}$ (light red dot), $\eta^2_{global}$ (light orange dot), and $\eta^2_{int,promoter}$ (light blue dot), mean SfGFP copy number (column of heatmap).

population are found in the stationary phase $\eta^2_{int,promoter}$ map (Fig. 1c and f, Fig. S4), indicating that more promoters are heterogeneously expressed in the stationary phase.

Due to the dependence of $\eta^2_{int,pure}$ on the mean protein expression level, promoters with low transcription rates exhibit high $\eta^2_{int,promoter}$, which is independent of the transcriptional regulation mode with which we are concerned. To estimate the expression noise brought about by transcriptional regulation, we used the residual of $\eta^2_{int,promoter}$ as a criterion for NTR, i.e., the vertical deviation of $\eta^2_{int,promoter}$ from the fitted curve ($\eta^2_{int,pure} = (b + 1)/{<}n{>}$, Fig. 1c).

$$\mathrm{NTR} = \eta^2_{int,promoter} - (b + 1)/{<}n{>} \tag{4}$$

$\eta^2_{int,pure}$ represents the theoretical intrinsic noise of a constitutive promoter with $n$ as the mean expression level. Genes within the same operon were assumed to have the same transcriptional divergence that is dependent on their shared promoter and therefore have the same NTR. Mean expression level, $\eta^2_{total}$, $\eta^2_{int,promoter}$, and NTR are presented on the whole-genome map, based on the chromosomal positions of their corresponding promoters (Fig. 1g and h). High values of mean expression level, $\eta^2_{total}$, $\eta^2_{int,promoter}$, and NTR are randomly distributed in the PAO1 genome (Fig. 1g and h, and see Table S2 at https://figshare.com/articles/dataset/Figures/21266778), although there are some locally large NTRs distributed around the chromosomal positions of 2.7, 3.8, and 4.6 Mbp in the stationary phase data (Fig. 1h).

**Promoters of genes belonging to different functional classes differed in their NTR levels.** To gain a detailed understanding of how specific functional categories relate to NTR, we grouped NTR data of reporters according to their functional classification (PseudoCAP) (22). Differences in median protein copy number (${<}G{>}$), $\eta^2_{int,promoter}$, or NTR belonging to different functional categories were estimated using single-sided Wilcoxon rank-sum test (Fig. 2a). The test was performed by taking the NTR values obtained from the stationary phase as an example and comparing all NTR values in a functional category of interest with the rest of the NTR values in the library. The same was done for the data from the exponential phase. In the exponential phase, relatively low levels of NTR are found in the categories of small-molecule transport, fatty acid and phospholipid metabolism, and cofactor biosynthesis. In contrast, relatively high levels of NTR are found in the categories of cell division, protein secretion, motility, and noncoding RNA expression. In the stationary phase, low levels of NTR are found in the categories of chaperone and heat shock proteins and transcriptional regulator, whereas high levels of NTR are found in the categories of chemotaxis and protein secretion. A previous study identified the core essential genome of the *P. aeruginosa* PA14 strain grown in LB (23). We mapped these essential genes in the PAO1 strain. In total, 320 genes driven by 196 promoters were screened out (see Table S3 at https://figshare.com/articles/dataset/Figures/21266778), accounting for 5.61% of all genes and 5.87% of all promoters. Except for the *PA1156* (*nrdA*) promoter, which has a moderate NTR (0.21) in the stationary phase (Fig. 2b and c), practically all of these promoters exhibit low NTRs in both growth phases (Fig. 2b and c), which is consistent with recent results in *Escherichia coli* (13). Essential genes are dose-sensitive; a low level of expression noise ensures that their dose never sinks too low. However, intrinsic stochasticity, contributed primarily by random transcriptional bursts, is inevitable. Through evolution, bacteria can acquire some regulatory mechanisms which lower the NTR, thus reducing total expression noise. However, we found few negative NTRs for those essential genes, suggesting that keeping their NTRs close to zero is enough to maintain bacterial fitness. In addition, a previous study demonstrated that proteins which respond to environmental changes are noisy and that the mode of transcription controls protein-specific noise (24). On the other hand, important low-NTR genes might be insensitive to environmental changes and constitutively expressed.

We next compared the data vertically across two growth phases; changes in NTR for genes within different functional categories appear to be independent of their corresponding changes in mean expression level. For example, the expression of genes belonging to the cell wall/lipopolysaccharide/capsule category is decreased in the

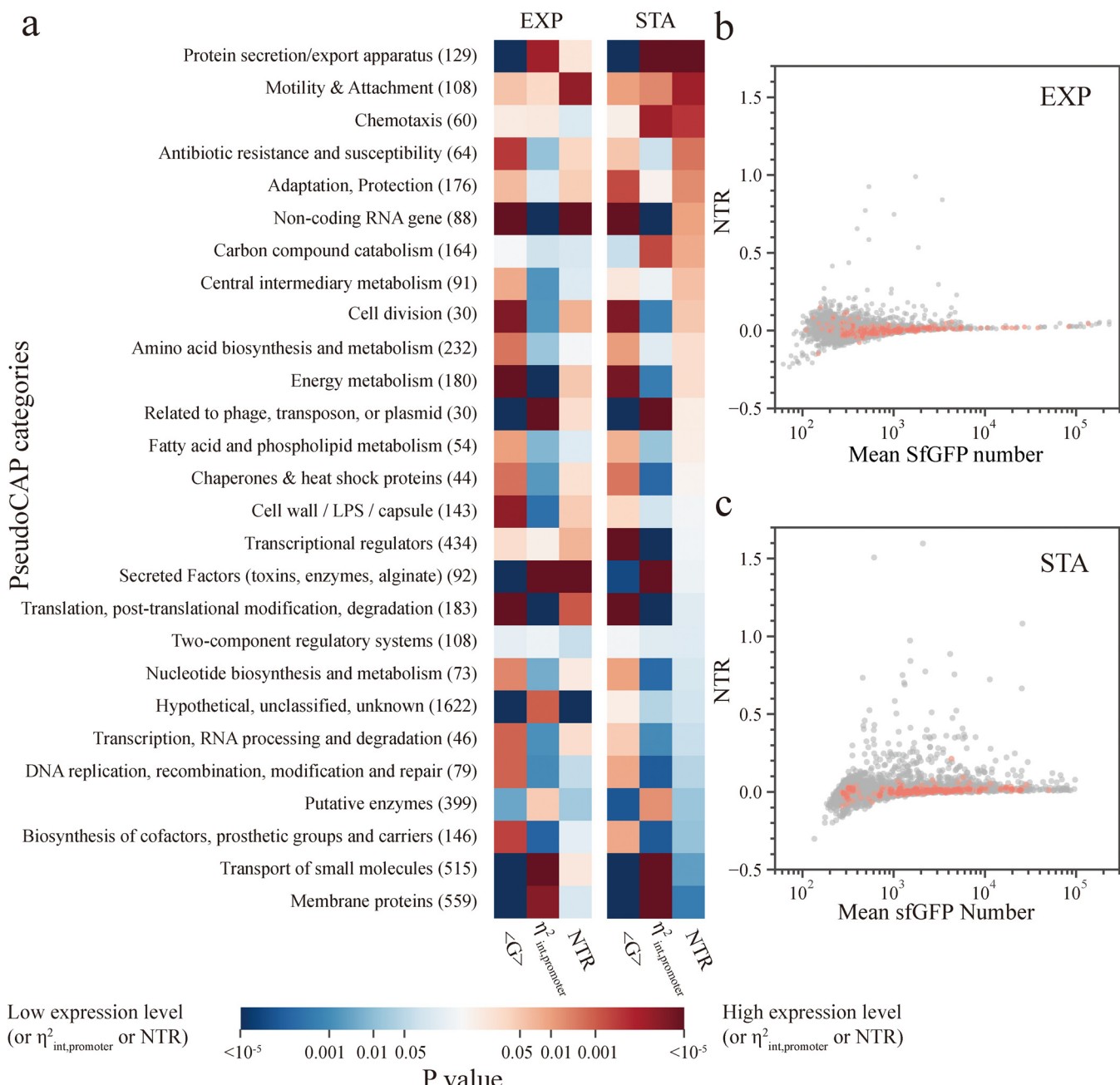

**FIG 2** Gene expression profiling based on functional categories. (a) Comparison of the mean SfGFP number, $\eta^2_{\text{int,promoter}}$, and NTR between promoter subgroups of 27 functional categories using a single-sided Wilcoxon rank-sum test. Numbers in brackets following functional category names represent total number of genes in the corresponding category. Test was conducted by comparing all the <G>, $\eta^2_{\text{int,promoter}}$, or NTR values in a functional category of interest with the rest of the values in the library. Red or blue colors indicate that the median <G> or NTR values of all reporters in the category of interest are larger (red) or smaller (blue) than the median <G> or NTR values in all other categories as one group. Bar color is based on the *P* value of the test. (b and c) Comparison of the NTR between promoters of core essential genes (orange dots) and other genes (gray dots) in PAO1.

stationary phase, whereas their NTRs are unchanged. Moreover, the expression levels of transcriptional regulators increases when cells enter the stationary phase, but their NTRs are relatively decreased (Fig. 2a). We further investigated this issue by visualizing the change of expression for each promoter against its corresponding NTR (Fig. S5). The correlation coefficients between NTR and mean protein expression change during the exponential and stationary phases are 0.24 and 0.18, respectively (Fig. S5a and b). The correlation coefficient between NTR change and mean protein expression change is 0.01 (Fig. S5c). These results indicate that there is no significant correlation between

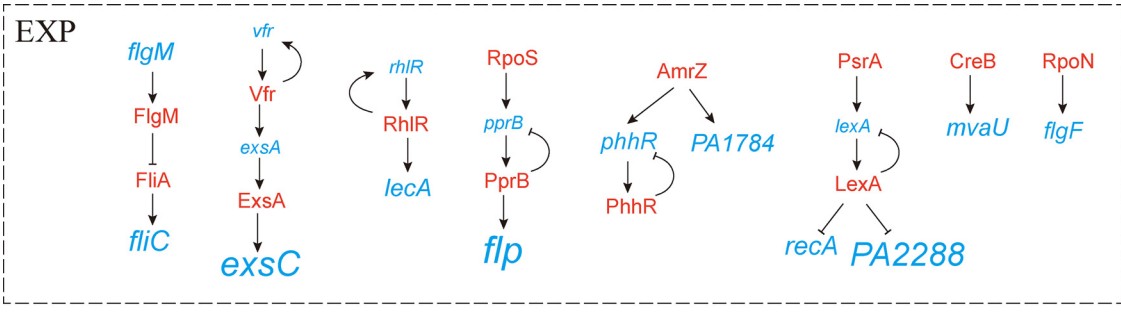

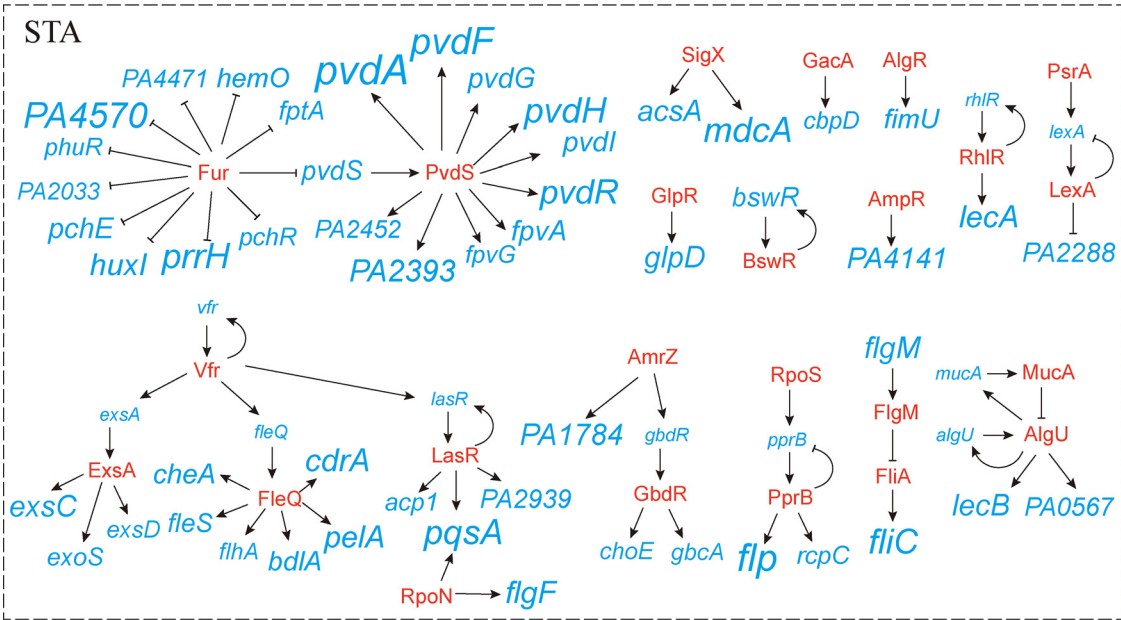

**FIG 3** Schematic illustration of currently known transcriptional regulatory networks that control the expression of high-NTR promoters in two growth phases. Transcriptional factors are presented in red text, promoters are presented in blue text and named after the first gene they control. Sizes of promoter names indicate the logarithmic value of their NTRs.

promoter NTRs and their promoter expression plasticity, which is consistent with a previous result in *E. coli* (12).

**Promoters with high NTR in *P. aeruginosa*.** We further used a fixed threshold (NTR = 0.15) to search for promoters with very high NTR, and a total of 140 promoters (287 genes) were screened out (Fig. S6, and see Table S4 at https://figshare.com/articles/dataset/Figures/21266778). The fixed NTR threshold of 0.15 is an empirical value. It was chosen for the following reasons. (i) According to our repeated experiments for 100 reporters, the experimental error for NTR measurement is less than 0.1 (Fig. S7a and b). Therefore, it is unlikely that an NTR value greater than 0.15 resulted from an experimental error. (ii) The 99% prediction bounds calculated through curve fitting seem to divide some data points within the measurement error into out-of-bound area (Fig. S7c and d). (iii) We hoped to screen those heterogeneously expressed promoters in an intuitive and certain manner; thus, floating thresholds depending on expression level were abandoned, and a larger fixed threshold was preferred. A fixed threshold also simplifies the assessment of gene expression heterogeneity in the following studies. (iv) We noticed a change in the NTR distribution curves (Fig. S7e) at NTR = 0.15, which might denote a change in the dominant mode of NTR genesis from experimental error to transcriptional regulation. We have provided a table of reporters whose NTR values exceeded the 99% upper prediction bounds (see Table S5 at https://figshare.com/articles/dataset/Figures/21266778).

The currently known transcriptional regulatory networks which control the expression of these promoters are shown in Fig. 3, based on data from the *Pseudomonas* database and previous studies (25–37).

Seventeen promoters (30 genes) showed high NTRs during both growth phases, including promoters of virulence genes involved in type III secretion system, type IVb pilus synthesis, and lectin production, and promoters of genes involved in transcriptional regulation (TR) and flagellum biogenesis. Notably, FlgM is an anti-sigma factor of FliA which directly controls the transcription of *fliC* promoter (29). NTRs of *flgM* and *fliC* promoters increase together when cells enter the stationary phase, suggesting that transcription noise was propagated from *flgM* to *fliC*.

Overall, 25 promoters (50 genes) showed high NTRs specific to the exponential phase. Among the proteins expressed by these promoters, BetI and PhhR are two transcriptional regulators controlling the expression of genes involved in glycine, phenylalanine, and tyrosine biosynthesis, and SpeC is an ornithine decarboxylase involved in arginine degradation (38–40). These results indicate the potential heterogeneity of cells in the metabolism of these amino acids during the exponential phase. In particular, an NTR of 0.84 was measured for the promoter of *PA5471.1*, which encodes a leader peptide that plays an inhibitory role in ArmZ activation, thereby inhibiting the expression of the downstream efflux pump MexXY-OprM (32). MexXY-OprM is involved in the efflux of multiple drugs; therefore, the heterogenous expression of *PA5471.1* may contribute to the persistence of *P. aeruginosa* in the presence of antimicrobials.

A total of 98 promoters (207 genes) showed high NTRs specific to the stationary phase. The corresponding genes have a wide range of functions, including chemotaxis, swimming motility, iron acquisition, biofilm formation, carbon metabolism, amino acid metabolism, and bacterial virulence. Interestingly, when bacterial cells enter the stationary phase, the NTRs of *phhR* and *betI* promoters are greatly reduced. In contrast, the NTRs of promoters of genes involved in glycine, lysine, serine, methionine, and histidine metabolism are increased, indicating a rearrangement of amino acid metabolism heterogeneity among the bacterial population. Moreover, large NTRs are also found in promoters of *pctA* and *pctB*, both of which encode methyl-accepting enzymes that are involved in bacterial chemotaxis toward multiple amino acids (41). Notably, a total of 21 promoters of genes or operons involved in iron acquisition exhibit high NTRs during the stationary phase, including genes encoding pyoverdine biosynthetic enzymes and transporters, pyochelin biosynthetic enzymes and outer membrane receptor, heme uptake receptor and heme oxygenase, and transcription factors for pyochelin and pyoverdine biosynthesis.

In *P. aeruginosa*, Fur is the central regulator for iron homeostasis that represses the transcription of iron uptake genes when bound to its corepressor, $Fe^{2+}$ (31, 42, 43). Under iron-limited conditions, transcriptional repression by Fur-$Fe^{2+}$ is relieved to enable the expression of certain iron uptake enzymes. According to previous studies, 10 of the high-NTR iron-related promoters (*pvdS*, *pchR*, *prrH*, *fptA*, etc.) are directly regulated by Fur (Fig. 3) (25–27, 31), suggesting a common Fur de-repression mechanism of noise generation, which is reminiscent of the peaking of transcriptional noise of a TetR-regulated gene under intermediate levels of tetracycline induction (44). In addition, the other 10 iron-related promoters (*pvdA*, *pvdF*, *pvdR*, *pvdI*, *fpvA*, etc.) with high NTRs are all within the direct regulon of sigma factor PvdS (Fig. 3) (28, 30), and most of them exhibit much larger NTRs than the *pvdS* promoter, indicating the amplification of *pvdS* transcriptional noise through PvdS-mediated transcriptional regulation. In addition, 6 promoters (*cheA*, *cdrA*, *pelA*, *bdlA*, *flhA*, *fleS*) are under the direct regulation of a c-di-GMP-dependent regulator, FleQ (Fig. 3) (34, 35), suggesting a common source of NTR.

**Heterogeneous gene transcription corresponds to heterogeneous bacterial physiological phenotype in pyoverdine production.** Given the role of post-transcriptional regulation (translational regulation, post-translational modification) within bacteria, heterogeneity at the transcription level does not necessarily correspond to the heterogeneity of protein expression and bacterial phenotype. To address this issue, we examined pyoverdine production at the single-cell level in *P. aeruginosa* and compared the data with the transcription levels of pyoverdine biogenesis genes. Pyoverdine efflux pump (*pvdR-pvdT-opmQ*) genes were deleted so that pyoverdine production could be estimated by measuring pyoverdine's characteristic fluorescence in single cells. As expected, there is great heterogeneity in pyoverdine production among *P. aeruginosa* cells (Fig. 4a to c). Moreover, the expression levels of promoters for most pyoverdine

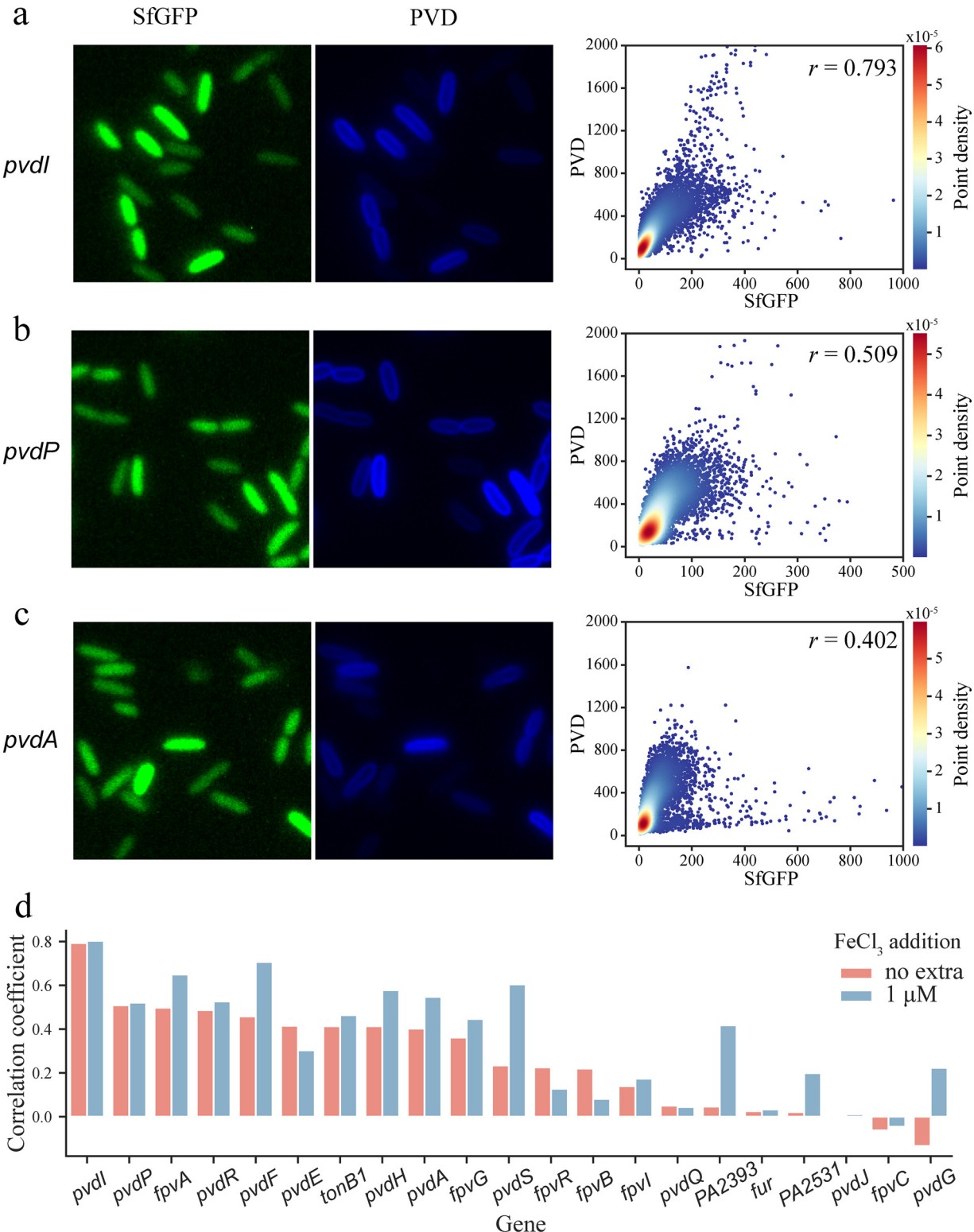

**FIG 4** Correlation analysis between the expression of pyoverdine (PVD) biogenesis genes and PVD production at the single-cell level. (a to c) Representative SfGFP, PVD fluorescence images of three reporter strains, with respective single-cell-PVD levels plotted against SfGFP copy numbers. (d) Correlation coefficient between single-cell SfGFP number and PVD fluorescence of multiple PVD-related reporters in FAB medium with or without FeCl$_3$ (1 $\mu$M) addition. $r$, correlation coefficient (Pearson).

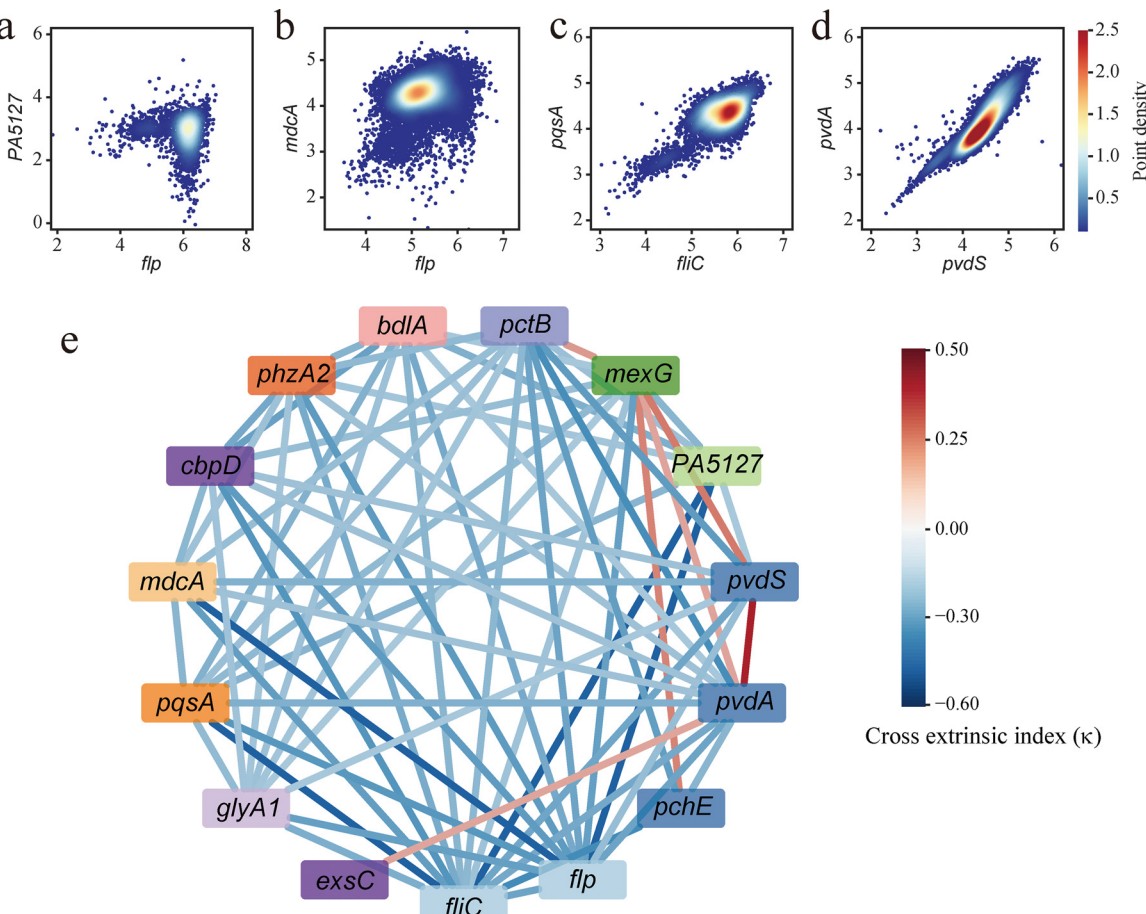

**FIG 5** Analysis of cross correlations between the transcription of promoters with high NTRs. (a to d) Representative expression patterns of dual-color reporters with SfGFP and CyOFP1 expression, respectively, driven by two high-NTR promoters. (a) Negative correlation between the transcription of *flp* and *PA5127* promoters. (b and c) Expression of *flp/mdcA* and *pqsA/fliC* reporters is distributed in relatively concentrated areas. (d) Linear positive correlation between the transcription of *pvdA* and *pvdS* promoters. (e) Cross-extrinsic noise index ($\kappa$) of dual-color cross reporters. $\kappa$ is the mean value of $\kappa_{p1,p2}$ and $\kappa_{p2,p1}$. Two promoters in mutual reporters with $\kappa > 0.25$ or $\kappa < -0.25$ were connected with red ($\kappa > 0.25$) or blue ($\kappa < -0.25$) line. Shade of red or blue lines indicates the $\kappa$ value according to the color bar. Background colors of promoters represents different functional categories to which they belong.

biosynthesis genes (*pvdA*, *pvdI*, *pvdP*, *pvdF*, *pvdH*) exhibit good positive correlations with cellular pyoverdine concentration (Fig. 4a to c), indicating a good correspondence of gene transcriptional heterogeneity with phenotypic heterogeneity in pyoverdine production. The transcription of *pvdI* shows the best correspondence with pyoverdine production (correlation coefficient $\approx$ 0.8) under both iron-deficient and iron-sufficient conditions (Fig. 4a). *pvdI* is a large gene encoding the 569.2-kDa non-ribosomal peptide synthetase, which provides a platform for pyoverdine assembly. This good correspondence suggests a rate-determining role of PvdI in pyoverdine production.

**Cross correlations of transcriptional regulation between high-NTR promoters.** To explore whether there are some inherent correlations between the transcription of those heterogeneously expressed promoters, we designed a new set of dual-color mutual reporters in which two target promoters drive the expression of *SfGFP* and *CyOFP1*. Fifteen promoters belonging to different functional categories were selected from the high-NTR promoters in the stationary phase, generating 105 ($C^2_{15}$) new dual-reporters. Distinct expression patterns were observed in these reporters (Fig. 5a to d, and see Fig. S9 at https://figshare.com/articles/figure/Supplementary_figures/21275937/2). For example, a negative correlation was observed between the expression of *flp* promoter and *PA5127* promoter, in which cells with lower levels of *flp* transcription generally

had higher levels of *PA5127* transcription (Fig. 5a). In contrast, the expression of *flp-mdcA* and *fliC-pqsA* reporters are distributed in relatively concentrated areas, indicating the low extrinsic noise of gene expression (Fig. 5b and c). In addition, the expression pattern of *pvdS-pvdA* reporter shows a positive linear correlation (Fig. 5d), reflecting extrinsic noise-dominated cell-to-cell variation in gene expression.

The expression levels of two independent promoters are usually positively correlated due to their shared extrinsic factors, such as RNA polymerase and ribosome. The contribution of these global factors to total expression noise is quantified by the extrinsic noise ($\eta^2_{\text{global}}$). To estimate the correlation of transcriptional regulation between two promoters, we defined the cross-extrinsic noise index as $\kappa_{\text{p1,p2}}$, which is determined by the following formula:

$$\kappa_{\text{p1,p2}} = \left( \eta^2_{\text{p1,p2,global}} - \eta^2_{\text{p1,global}} \right) / \eta^2_{\text{p1,global}} \tag{5}$$

where $\eta^2_{\text{p1,p2,global}}$ is the extrinsic noise (calculated according to equation 2) of the mutual reporter with promoters *p1* and *p2*. This $\eta^2_{\text{p1,p2,global}}$ quantifies the contribution of total expression noise from common factors that control both *p1* and *p2* expression, including their common transcriptional factors and mutual regulations. $\eta^2_{\text{p1,p2,global}}$ is the extrinsic noise obtained from the previously mentioned transcriptional reporter library (Fig. 1e); it quantifies the contribution of only global factors that control *p1* and J23102 expression. $\kappa_{\text{p1,p2}}$ is the estimation of the amplification or reduction of extrinsic noise due to the correlation of transcriptional regulation between *p1* and *p2*, which is independent of global factors such as levels of RNA polymerase or ribosome. $\kappa_{\text{p1,p2}} < 0$ indicates a negative correlation between *p1* and *p2* transcriptional regulation that reduces the extrinsic noise, while $\kappa_{\text{p1,p2}} > 0$ indicates a positive correlation between *p1* and *p2* transcriptional regulation that amplifies the extrinsic noise. $\kappa_{\text{p1,p2}} = 0$ indicates no correlation between *p1* and *p2* transcriptional regulation, as is the case for *p1* to a constitutive promoter.

In general, the $\kappa_{\text{p1,p2}}$ of most mutual reporters are negative (Fig. 5e, and see Table S6 at https://figshare.com/articles/dataset/Figures/21266778), indicating a global negative correlation of transcriptional regulation between heterogeneously expressed genes. It is unclear what the underlying mechanics are. Bacteria may maintain subpopulations which employ various survival strategies while under stress.

The most negative $\kappa_{\text{p1,p2}}$ are seen in *flp-PA5127* (−0.82) and *flp-PA0208* (−0.69) reporters (Fig. 5e), in which *flp* encodes the type IVb pilin, *PA5127* encodes an rRNA methylase, and the *PA0208* promoter drives the expression of several proteins comprising the malonate decarboxylase complex (see the *Pseudomonas* database at https://www.pseudomonas.com). Of the few reporters with positive $\kappa_{\text{p1,p2}}$, the *pvdS-pvdA* reporter exhibits the most positive $\kappa_{\text{p1,p2}}$, which is expected because PvdS controls *pvdA* transcription. In addition, the *mexG-pctB*, *mexG-pchE*, and *mexG-pvdS* reporters exhibit relatively large positive $\kappa_{\text{p1,p2}}$, suggesting that there is some functional synergy between the MexGHI-OpmD efflux pump and chemotaxis system and pyochelin/pyoverdine biosynthesis in the stationary phase. Notably, transcriptional regulation of *fliC* promoter exhibits negative correlations ($\kappa_{\text{p1,p2}} < 0$) with all other 14 promoters (Fig. 5e), indicating that the biosynthesis of flagella is mutually exclusive with other functional categories in response to environmental stress. In short, when bacteria decide to swim away, they may no longer require the proteins that enable them to survive in harsh environments.

**Curve fitting of SfGFP distribution curves with gamma and lognormal distribution.** According to previous studies, variation in protein expression follows a gamma distribution when stochastic bursting dominates other sources of intracellular variation (14, 45), which is usually the case for constitutive promoters with a low transcription rate. When the transcription rate is high so that stochastic bursting does not dominate, many complex biochemical reactions, such as sigma factor binding, transcriptional elongation, promoter escape, and tRNA-amino acid binding, lead to deviation of the protein expression process from the simple model (45). The latter situation often

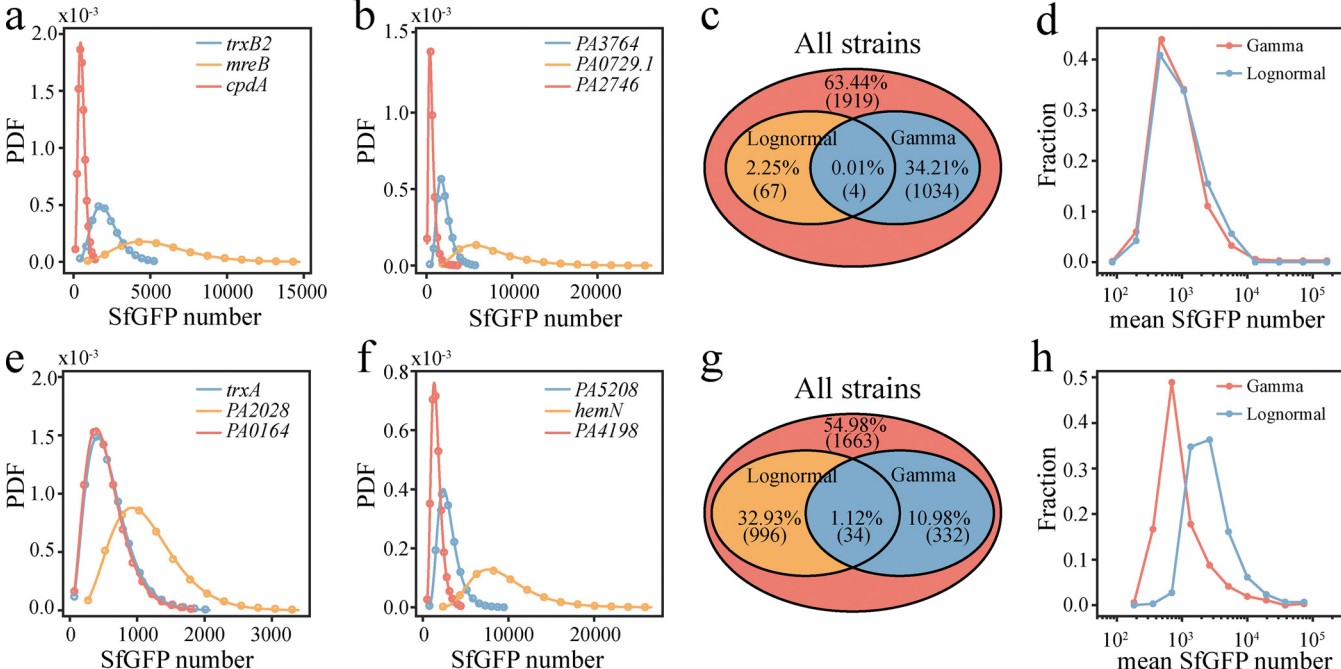

**FIG 6** Log-normal and gamma fit of single-cell-SfGFP number distributions. (a to d) Data for the exponential phase. (e to h) Data for the stationary phase. (a, b, e, f) Representative SfGFP distributions of three reporters which fit well with gamma distribution (a, e) or log-normal distribution (b, f). PDF, probability distribution function. (c, g) Venn diagrams representing the proportion and number of reporters whose SfGFP distributions fit well with gamma or log-normal distribution. (d, h) Mean SfGFP expression histograms of reporters whose SfGFP distributions fit well with gamma (light red) or lognormal (light blue) distribution.

results in a log-normal distribution of protein expression (46). To address this issue, we fitted the distribution curves of SfGFP for all reporters using both the gamma distribution and the log-normal distribution. In the exponential growth phase, the data of 1,038 reporters can be fitted ($P > 0.05$) with gamma distribution (Fig. 6a and c), whereas those of only 71 reporters can be fitted with log-normal distribution (Fig. 6b and c). In the stationary growth phase, the data of 366 reporters can be fitted with gamma distribution, and those of 1,030 reporters can be fitted with log-normal distribution (Fig. 6e to g). However, lower protein expression of gamma-distributed reporters is only found during the stationary phase (Fig. 6d and h). In addition, the NTRs of those well-fitted (gamma or log-normal) promoters are similar to promoters whose cell-to-cell expression cannot be fitted with simple distributions (Fig. S8), indicating a considerable degree of transcriptional regulation in some well-fitted promoters.

**High-NTR reporters exhibit two distinct expression patterns.** We found two distinct expression patterns of high-NTR reporters. Promoters of *pvdP*, *prrH*, and *PA5127* exhibit bimodal expression patterns (Fig. 7a and b), while those of *metE*, *PA4121*, and *cdrA* exhibit a unimodal one (Fig. 7c and d). The largest bimodality coefficient (47) of SfGFP/CyOFP1 distributions was observed in promoters of *PA5127* (0.70), *exsC* (0.69), *PA4580* (0.67), and *flp* (0.66) in the exponential phase and in promoters of *pvdA* (0.79), *PA4073* (0.78), *pvdF* (0.75), and *PA4570* (0.73) in the stationary phase (see Table S2 at https://figshare.com/articles/dataset/Figures/21266778). The bimodal expression patterns of these promoters indicate probable phenotypic separation in the expression of these genes.

## DISCUSSION

We have conducted a genome-wide analysis of NTR for *P. aeruginosa* promoters. Separation of the promoter-specific intrinsic noise from the total noise with a dual-color reporter facilitates the characterization of gene expression variability brought about by transcriptional regulation, as measured by NTR. Interestingly, we found high NTRs in a wide range of promoters driving the expression of iron-acquisition genes during the

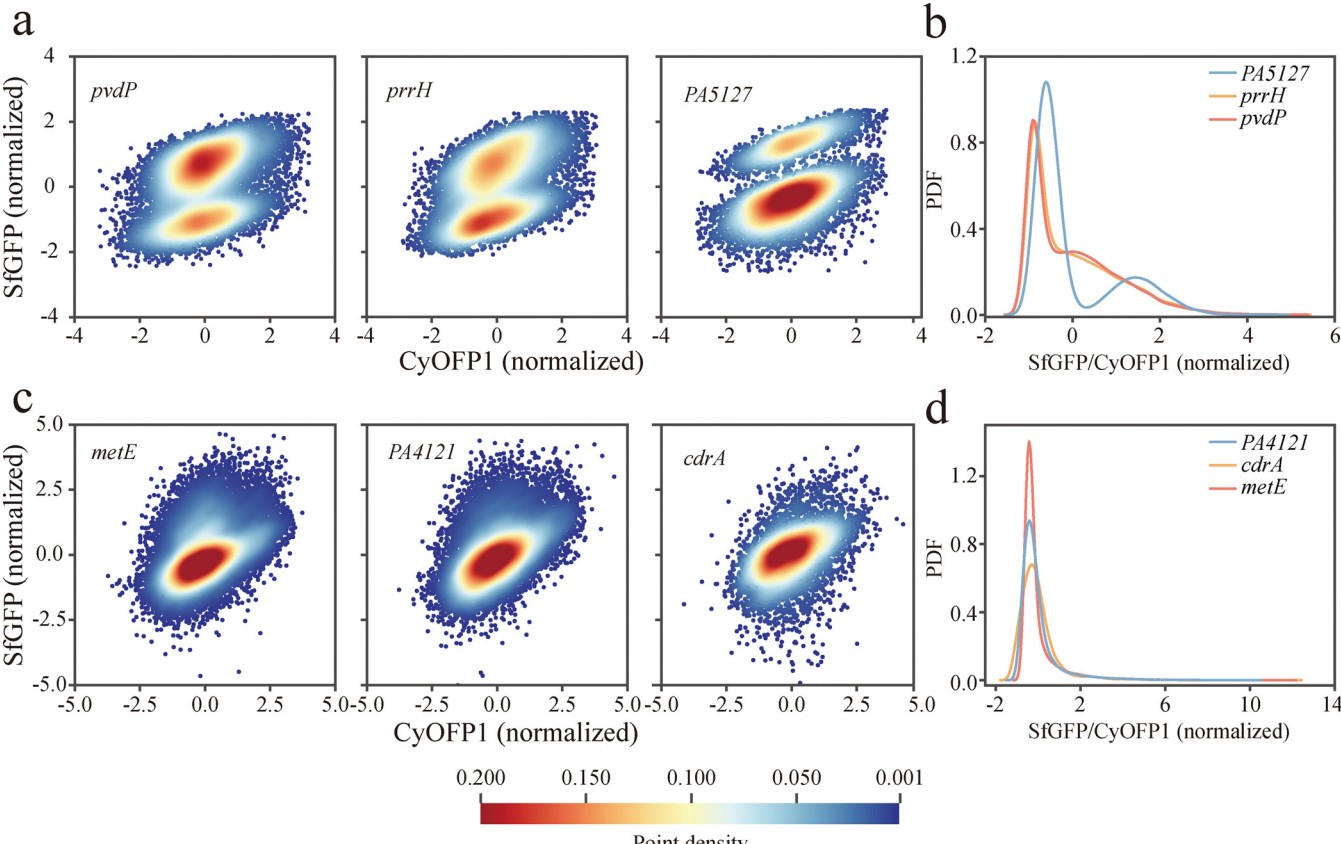

**FIG 7** Two expression patterns of high-NTR reporters. (a, b) Representative bimodal expression patterns of *pvdP*, *prrH*, and *PA5127* promoters (a) and their corresponding SfGFP/CyOFP1 distributions (b). (c, d) Representative unimodal expression patterns of *metE*, *PA4121*, *cdrA* promoters (c) and their corresponding SfGFP/CyOFP1 distributions (d). The expression of SfGFP and CyOFP1 in the heat map were taken from the logarithm and normalized by Z-score.

stationary growth phase, and a large-scale negative dependence of transcriptional regulation was also found between high-NTR promoters belonging to different functional categories. Genome-wide studies of NTR under other usual growth circumstances, such as host infection or antibiotic stress, are necessary to acquire more knowledge for clinical use. Additionally, while some putative explanations for the high NTRs of the promoters we tested have been postulated, the underlying molecular pathways remain unknown. To obtain further insights into issues such as how transcription noise propagates through the iron-Fur-PvdS-*pvdAFGHPRI* regulation cascade and why the *pvdA* promoter exhibits significantly higher NTR than promoters of other pyoverdine synthesis genes will likely require the clarification of detailed binding dynamics of transcription factors, with a combination of biophysical modeling.

It should be noted that the PUCP20 vector used in this study has a moderate copy number in *P. aeruginosa*. Thus, the mean expression levels of the promoters we measured were higher than the corresponding chromosomal promoters, which enhances our system's detection limit and enables measurement of low-expressed promoters. Variation in the plasmid copy number contributes to the extrinsic noise of gene expression, whereas multiple identical copies of promoter-gene pairs in a cell reduces the pure intrinsic noise $\eta^2_{\text{int,pure}}$ ($\eta^2_{\text{int,pure}} = \eta^2_{\text{int,promoter}} - \text{NTR}$) by the scheme $\eta^2_{\text{int,pure}} \propto 1/N$, where $N$ is the plasmid copy number (48). Notably, the NTR is mainly dependent upon the variation of copy numbers of the transcription factor which contributes to an extrinsic factor of the expression of parallel promoter-gene pairs, thus having minimal dependence on gene copy numbers (48). Therefore, our measured results of extrinsic noise are larger than that in actual chromosome-expressed genes, while the measured intrinsic noise is lower than that in the real chromosomal context. However, our results on $\eta^2_{\text{int,promoter}}$ and NTR are still reasonable for the horizontal comparison of transcriptional heterogeneity for different promoters,

which is supported by the expected reciprocal relationship between $\eta^2_{\text{int,promoter}}$ and protein copy number.

In addition to measuring fluorescent transcriptional reporters, single-cell RNA sequencing (scRNA-Seq) (49–51) can also provide information about gene expression heterogeneity. scRNA-Seq would provide parallel transcription data for a vast number of promoters in thousands of single cells and facilitate the clustering of subpopulations in the bacterial community. However, a previous study demonstrated that a single-cell's protein and mRNA copy numbers are not correlated for any given gene, probably due to the distinct lifetime between mRNA and protein and the extrinsic translational noise between cells. In addition, the low mRNA copy number in bacterial cells often presents difficulties in distinguishing gene expression noise from experimental error. On the other hand, fluorescent transcriptional reporters enable the correlation analysis of gene expression with bacterial physiological phenotypes (such as the kill rate of antibiotics, single-cell growth rate, and siderophore production) *in situ* under a microscope. Therefore, constructing transcriptional reporters is still an important method for studying gene expression noise, and cannot be replaced by scRNA-Seq.

In this study, we measured the NTR of transcriptional reporters in LB broth, which is the most commonly used medium in the laboratory. However, the natural habitats of *P. aeruginosa* are soil, water, vegetation, and animal hosts. The primary concerns for this bacterium are how to deal with its infection in human wounds and lungs and how to clear its biofilms, due to the rising challenge of antibiotic failure. On the other hand, with changes in growth conditions involving carbon sources, nitrogen sources, oxygen concentration, temperature, and antibiotic stress, both the expression level and activation state of transcription factors change in *P. aeruginosa*. Therefore, our results in LB represent only two examples; the NTR of bacteria may change substantially under different growth conditions. To gain further insights into NTR regarding bacterial physiology in a biofilm, under antibiotic stress, and in infection, experiments using minimal media with single controllable variables are needed.

The large NTRs of iron-acquisition genes indicate a great diversity of iron physiology in the bacterial population. Iron is essential for the tricarboxylic acid cycle and bacterial respiration via its incorporation into key enzymes in iron-sulfur clusters or heme groups, while *in vivo* ferrous ion also mediates the amplification of reactive oxygen species attack through the Fenton reaction (52). These opposite effects may prompt the bacteria to adopt a bet-hedging strategy to control intracellular iron concentrations under environmental stress. The hypothesis derived from NTR measurements awaits future experimental testing.

## MATERIALS AND METHODS

**Bacterial strains and growth conditions.** All strains used in this study are based on the *Pseudomonas aeruginosa* strain PAO1. The transcriptional reporter strains used in this study are listed in Table S2 (https://figshare.com/articles/dataset/Figures/21266778). Unless otherwise stated, 30 $\mu$g/mL gentamicin was added to all culture media to prevent plasmid loss. All bacterial culture experiments were conducted at 37°C. For transcriptional noise measurements, each cryopreserved ($-80$°C) reporter strain was recovered on an LB agar plate and incubated for 12 h. Next, bacterial colonies were scraped into a 5.5-mL polystyrene tube containing 1 mL LB broth medium and cultured in a shaker for 12 h to reach stationary growth phase. The bacterial culture was further diluted 500-fold in 1 mL of fresh LB medium and allowed to grow to mid-exponential phase (optical density at 600 nm [OD$_{600}$] $\approx$ 0.6) for another 3.5 h and shaken at 220 rpm. Cells in the stationary and exponential phases were diluted 100-fold and 10-fold in transparent fastidious anaerobe broth (FAB) medium (see supplementary material for medium composition) (53), respectively. Next, 40 $\mu$L of the diluted cultures was suctioned into a glass-bottomed dish for microscopic observation.

**Validation of the reporter library.** For all culture media, 30 $\mu$g/mL gentamicin was added. In the ciprofloxacin treatment experiment, reporter strains of *lexA*, *dinB*, *PA2288*, *pyoS5*, *pys2*, and *PA3866* were cultured into the exponential growth phase in FAB medium containing 30 mM glutamate. One $\mu$g/mL of ciprofloxacin was added to the bacterial cultures, and they were grown for 4 h. Bacteria were harvested for microscope observation. For the carbon-source treatment experiment, reporter strains of *aceA* and *acsA* were grown overnight in FAB medium containing 30 mM acetate or 30 mM succinate as the sole carbon source. The cultures were then diluted 100 times in the same medium and grown to the exponential phase for the observation of *cupA1* reporter in the wild-type and *mvaT* mutant strains. Bacterial strains were grown in LB overnight and diluted 100 times until grown into

 

the exponential phase. FAB minimal medium was composed as follows: ammonium sulfate [(NH$_4$)$_2$SO$_4$; 2 g/L], disodium hydrogen phosphate dodecahydrate (Na$_2$HPO$_4$; 2 H$_2$O, 12 g/L), potassium dihydrogen phosphate (KH$_2$PO$_4$; 3 g/L), sodium chloride (NaCl; 3 g/L), magnesium chloride (MgCl$_2$; 93 mg/L), calcium chloride (CaCl$_2$; 14 mg/L), ferric chloride (FeCl$_3$; 1 $\mu$M), and trace metals solution (1 mL/L). The trace metals solution was composed as follows: calcium sulfate dihydrate (CaSO$_4$; 2 H$_2$O, 200 mg/L), manganese sulfate monohydrate (MnSO$_4$; 2 H$_2$O, 200 mg/L), copper sulfate pentahydrate (CuSO4; 5 H$_2$O, 20 mg/L), zinc sulfate heptahydrate (ZnSO$_4$; 7 H$_2$O, 20 mg/L), cobalt sulfate heptahydrate (CoSO$_4$; 7 H$_2$O, 10 mg/L), sodium molybdate monohydrate (Na$_2$MO$_4$; H$_2$O, 10 mg/L), and boracic acid (H$_3$BO$_3$; 5 mg/L).

**Construction of transcriptional reporters.** A general template plasmid, PUCPgfp, was constructed, in which *sfGFP*, *cyOFP1*, and terminator fragments were amplified and cloned into the PUCP20 backbone via Gibson assembly. The resultant genetic organization was *RNAseIII*-RBS2-*sfGFP*-T$_0$T$_1$-J23102-RBS2-*cyOFP1*-T-PUCP20, in which J23102 is a constitutive promoter (http://parts.igem.org/Part:BBa_J23102). Promoter fragments of all ORFs in PAO1 were amplified using primers designed by a self-written code based on MATLAB Bioinformatics Toolbox. The captured promoter sequences consisted of the whole intergenic region as well as 0 to 300 bp upstream genes and 0 to 100 bp downstream genes. The promoter fragments were separately cloned into PUCPgfp immediately in front of the *RNAseIII* site, generating the transcriptional reporter plasmid library. Those plasmids were extracted from the top 10 strains and introduced by chemical transformation into the *P. aeruginosa* PAO1 strain. High-efficiency PAO1 competent cells were prepared using an Ultra-Competent Cell Preps kit (Sangon Biotech, Shanghai, China), according to the manufacturer's instructions. To construct dual-color reporter plasmids for the purpose of studying the cross-correlation of two distinct promoters, the J23102 promoter from the original *exsC* (or *pqsA*, *glyA1*, etc.) reporter plasmid was substituted with the desired promoter. Plasmids and strains are available on request.

**Fluorescent image acquisition and data analysis.** Fluorescent images of SfGFP and CyOFP1 were acquired simultaneously using two Zyla 4.2 sCMOS cameras (Andor) on a fluorescence microscope (IX-71, Olympus) equipped with a 100× oil objective. The fluorescence of both SfGFP and CyOFP1 was excited at 488 nm using a solid-state light source (Lumencor Spectra X), and collected with emission filters of 520/28 nm and 583/22 nm, respectively. For exponential-phase and stationary-phase bacterial samples, 5 × 5 and 4 × 4 montage images were collected, respectively. Both cameras have a visual field of 2,048 × 2,048 pixels, equivalent to 133 × 133 $\mu$m when the 100× oil objective was used. Each field contained 200 to 400 single cells for the exponential-phase samples and 800 to 1,000 single cells for stationary-phase samples. Thus, for each sample, gene expression data of approximately 10,000 single cells were obtained. Image processing was conducted using MATLAB with a self-written code. SfGFP and CyOFP1 images were aligned using a built-in function (imwarp) and CyOFP1 images were used for bacterial cell identification. To identify cell masks, CyOFP1 images were subsequently filtered with a gaussian filter, edge filter, and gaussian filter using a built-in function (imfilter). The images were then transformed into binary images using a built-in function (im2bw) with a fixed intensity threshold (see Fig. S10 at https://figshare.com/articles/figure/Supplementary_figures/21275937/2). These binary images were called mask images because the connected regions in them indicate the occupied areas of bacterial cells in CyOFP1 images. Single-cell intensities of SfGFP and CyOFP1 were determined from mask images using a built-in function (regionprops). A fixed pixel threshold of the bacterial occupied area was used so that only the intensities of separate single cells were extracted for data collection.

The cross talk correction of SfGFP and CyOFP1 intensities and the transformation of fluorescent intensity to protein concentration were performed in accordance with a previous study (54). For a single cell containing both SfGFP and CyOFP1, the obtained fluorescent intensity can be calculated as follows:

$$(I_{\text{image,G}}, I_{\text{image,O}}) = (I_{\text{real,G}}, I_{\text{real,O}}) \begin{pmatrix} 1 & \beta_{GO} \\ \beta_{OG} & 1 \end{pmatrix} \tag{6}$$

where $I_{\text{image,G}}$ and $I_{\text{image,O}}$ are the SfGFP and CyOFP1 intensities obtained from original images, and $I_{\text{real,G}}$ and $I_{\text{real,O}}$ are the expected SfGFP and CyOFP1 intensities of the cell when there is no cross-talk between SfGFP and CyOFP1. $\beta_{GO}$ and $\beta_{OG}$ are SfGFP-to-CyOFP1 and CyOFP1-to-SfGFP cross-talk coefficients, respectively. Equation 6 can be written in the vector form as:

$$I_{\text{image}} = I_{\text{real}}\beta \tag{7}$$

Then, the expected SfGFP and CyOFP1 intensities can be calculated as follows:

$$I_{\text{real}} = I_{\text{image}}\beta^{-1} \tag{8}$$

where $\beta^{-1}$ is the inverse matrix of $\beta$, called the cross-talk matrix. $\beta_{GO}$ and $\beta_{OG}$ were obtained as follows (see Fig. S11, Fig. S12 at https://figshare.com/articles/figure/Supplementary_figures/21275937/2). PAO1 strains expressing only SfGFP (PAO1::*PA1O4O3-sfGFP*-miniTn7, represented as PA_gfp strain) or CyOFP (PAO1::*PA1O4O3-cyOFP*-miniTn7, represented as PA_ofp strain) were constructed first. Bacterial cells were cultured in LB, resuspended in FAB medium, and sandwiched between a thin 2% agarose pad and a glass coverslip on the bottom. SfGFP and CyOFP images were captured for both the PA_gfp and PA_ofp strains. Then, single-cell intensities of SfGFP and CyOFP for PA_gfp and PA_ofp were calculated following the procedure described above (the mask images of PA_gfp strain were calculated from SfGFP images). $\beta_{GO}$ and $\beta_{OG}$ were obtained according to the linear regression of the scatter diagram of SfGFP-to-CyOFP1 intensity for PA_gfp and PA_ofp (see Fig. S11 at https://figshare.com/articles/figure/Supplementary_figures/21275937/2).

The protein concentrations of SfGFP and CyOFP1 in single cells were determined according to the concentration-intensity relationship of purified fluorescent proteins. Briefly, SfGFP and CyOFP1 proteins were purified using the Ni-chelating affinity chromatography method. Purified SfGFP and CyOFP1 protein solutions were diluted to a fixed concentration, and silica microspheres of different sizes (0.5 to 10 $\mu$m) were added. Then, each solution was added in between two 18 mm $\times$ 18 mm cover glasses. Cover glasses were firmly pressed to squeeze out excess solution, and the edges were sealed with jelly petroleum. The solution layer thickness of each sample was based on the size of added silica microspheres. Then, the fluorescence of these samples was measured and the working curve of fluorescence against solution layer thickness was obtained. A solution layer thickness of 0.5 $\mu$m was picked as the thickness of a bacterial cell lying flat. The working curve of fluorescence against different protein concentrations was obtained using the same method. Then, the intracellular SfGFP and CyOFP1 fluorescence intensities of single cells were transformed into protein concentrations according to these two work curves. Protein numbers in single cells were calculated by multiplying protein concentration with the estimated cell volume (2/3 $\times$ cell occupied area $\times$ cell width).

**Functional classification of genes in *P. aeruginosa*.** Functional classification of genes was based on the *Pseudomonas* Community Annotation Project (PseudoCAP) data from the *Pseudomonas* database (https://www.pseudomonas.com/). Differences in intrinsic noise, NTR, and mean expression level of genes belonging to different functional categories were estimated using a single-sided Wilcoxon rank-sum test (MATLAB built-in function, ranksum). Essential genes of the PAO1 strain grown in LB medium were determined according to the methods of a previous study (23), and gene details are listed in Table S3 ( https://figshare.com/articles/dataset/Figures/21266778).

**Single-cell measurement of pyoverdine production.** Pyoverdine fluorescence was measured in cells devoid of the *pvdR-pvdT-opmQ* operon. Markerless gene knockout was performed according to a previous protocol (55). Briefly, 1,000-bp upstream and downstream sequences for *pvdR-pvdT-opmQ* operon were amplified from the PAO1 genome and cloned into plasmid pex18gm via Gibson assembly. The resulting plasmid was extracted, electroporated into PAO1, and plated on an LB agar plate supplemented with 30 $\mu$g/mL gentamicin. Single colonies were streaked on a 15% sucrose no-salt LB agar plate, and the resultant inoculums were selected by PCR verification and sequencing to obtain the final *pvdR-pvdT-opmQ* knockout strain. Transcriptional reporter plasmids of interest were electroporated into this strain, generating reporter strains for the study of pyoverdine production in relation to gene expression. The reporter strains were grown overnight in LB medium, washed twice with FAB medium, and resuspended in FAB. Cell suspensions were diluted 500$\times$ in FAB medium supplemented with 30 mM sodium glutamate, with or without the addition of 1 $\mu$M FeCl$_3$. After 6 h of shaking at 220 rpm and 37°C, cells were harvested and pipetted into a glass-bottomed dish for microscope imaging. Pyoverdine fluorescence was excited at 420 nm and emission was measured at 464/30 nm using the same camera as used for SfGFP image collection. Pyoverdine intensities of single cells were also determined using the mask generated from CyOFP1 images.

**Data availability.** Original gene expression data for all transcriptional reporters are included in the supplemental material. MATLAB codes for promoter design and fluorescent image processing have been uploaded in GitHub (https://github.com/Carl-Ni/promoterDesign_singleCellIntensityExtraction_Code).

## SUPPLEMENTAL MATERIAL

Supplemental material is available online only.
**FIG S1**, TIF file, 3.2 MB.
**FIG S2**, TIF file, 1.5 MB.
**FIG S3**, TIF file, 9.1 MB.
**FIG S4**, TIF file, 1.2 MB.
**FIG S5**, TIF file, 1.1 MB.
**FIG S6**, TIF file, 0.6 MB.
**FIG S7**, TIF file, 2.4 MB.
**FIG S8**, TIF file, 1.4 MB.

## ACKNOWLEDGMENTS

We thank Qing Wei for his constructive advice on constructing the transcriptional library of *P. aeruginosa*. We thank Xiao Yi for his kind advice on the presentation of experimental data.

This work was supported by the National Key Research and Development Program of China (grant no. 2020YFA0906900 and grant no. 2018YFA0902700 to F.J.), the National Natural Science Foundation of China (grant no. 32000060 to A.X.) and the Scientific Instrument Developing Project of the Chinese Academy of Sciences (grant no. YJKYYQ20200033 to F.J.).

Conceptualization, L.N. and F.J.; transcriptional reporter library construction, W.C., J.Z., C.W., Y.Z., F.L., L.N.; image data acquisition, A.X., W.C., J.Z., F.L.; data analysis, W.C.; investigation, W.C., L.N., F.J.; writing – original draft, L.N., writing – review and editing, F.J.

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
