## [Reviewer comments · mSystems]

Genome-wide analysis of gene expression noise brought about by transcriptional regulation in *Pseudomonas aeruginosa*

Wen Chen, JinFeng Zhang, Feixuan Li, Cong Wang, Yuchen Zhang, Aiguo Xia, Lei Ni, and Fan Jin

Corresponding Author(s): Fan Jin, Shenzhen Institute of Synthetic, Shenzhen Institutes of Advanced Technology Chinese Academy of Sciences

Review Timeline:

Submission Date:

October 6, 2022

Accepted:

October 25, 2022

Editor: Alejandra Rodríguez-Verdugo

Reviewer(s): The reviewers have opted to remain anonymous.

Transaction Report:

DOI: <https://doi.org/10.1128/msystems.00963-22>

October 25, 2022

Dr. Fan Jin
Shenzhen Institute of Synthetic, Shenzhen Institutes of Advanced Technology Chinese Academy of Sciences
Shenzhen
China

Re: mSystems00963-22 (Genome-wide analysis of gene expression noise brought about by transcriptional regulation in *Pseudomonas aeruginosa*)

Dear Dr. Fan Jin:

Thank you for re-submitting your manuscript and for responding point-by-point to the reviewer's comments. I have served as a reviewer of your manuscript. Previously, we all agreed that this study addressed a relevant question, and it was scientifically sound. The concerns revolved around the presentation of the manuscript and the data interpretation. I am happy to say that, after addressing the previous reviewers' critique, the manuscript has greatly improved, and it is acceptable in its current form. I would just advise addressing the following two minor points: in lines 363-365, please change "It's" to "It is" and add a reference; and in line 380, please change "Put simply" to "In short". Thank you again for re-submitting your work and addressing all the reviewers' comments.

Your manuscript has been accepted, and I am forwarding it to the ASM Journals Department for publication. For your reference, ASM Journals' address is given below. Before it can be scheduled for publication, your manuscript will be checked by the mSystems production staff to make sure that all elements meet the technical requirements for publication. They will contact you if anything needs to be revised before copyediting and production can begin. Otherwise, you will be notified when your proofs are ready to be viewed.

Publication Fees:

If you would like to submit a potential Featured Image, please email a file and a short legend to mssystems@asmusa.org. Please note that we can only consider images that (i) the authors created or own and (ii) have not been previously published. By submitting, you agree that the image can be used under the same terms as the published article. File requirements: square dimensions (4" x 4"), 300 dpi resolution, RGB colorspace, TIF file format.

We recognize that the video files can become quite large, and so to avoid quality loss ASM suggests sending the video file via

<https://www.wetransfer.com/>. When you have a final version of the video and the still ready to share, please send it to mSystems staff at mSystems@asmusa.org.

Sincerely,

Alejandra Rodríguez-Verdugo
Editor, mSystems

Journals Department
Fig. S4: Accept

Fig. S7: Accept

Fig. S2: Accept

Fig. S5: Accept

Fig. S8: Accept

Fig. S3: Accept

Fig. S1: Accept

Fig. S6: Accept